# High-Level Expression of Recombinant VHSV Glycoprotein Using Transformed *C. vulgaris* and Verification of Vaccine Efficacy

**DOI:** 10.3390/vaccines11071205

**Published:** 2023-07-05

**Authors:** Min-Jeong Kim, Seon-Young Kim, Ki-Hong Kim, Sung-Sik Yoo, Taek-Kyun Lee, Tae-Jin Choi

**Affiliations:** 1Department of Microbiology, School of Marine and Fisheries Sciences, Pukyong National University, Busan 48513, Republic of Korea; rnfma00082@pukyong.ac.kr; 2Department of Aquatic Life Medicine, Pukyong National University, Busan 48513, Republic of Koreakhkim@pknu.ac.kr (K.-H.K.); 3Choong Ang Vaccine Laboratory Co., Ltd., South Chungcheong, Daejeon 34055, Republic of Korea; 4South Sea Environment Research Division, Korea Institute of Ocean Science & Technology, Geoje-si 53201, Republic of Korea

**Keywords:** VHSV, oral vaccine, chlorella, microalga, salt-inducible promoter

## Abstract

The demand for aquaculture is increasing, but production is declining due to high feed costs and disease outbreaks. Viral hemorrhagic septicemia (VHS) is a viral disease that seriously affects seawater and freshwater fish in aquaculture, including the olive flounder (*Paralichthys olivaceus*), a major aquaculture fish in Korea. However, very few vaccines are currently available for viral hemorrhagic septicemia virus (VHSV). The nutrient-rich microalga *Chlorella vulgaris* has been used as a feed additive in aquaculture and as a host for the industrial production of recombinant VHSV glycoprotein as an oral vaccine. In this study, VHSV glycoprotein was cloned with a salt-inducible promoter, and high levels of expression up to 41.1 mg/g wet *C*. *vulgaris*, representing 27.4% of total extracted soluble protein, were achieved by growing the transformed *C*. *vulgaris* for 5 days in the presence of 250 mM NaCl. The production of a neutralizing antibody was detected in the serum of fish given feed containing 9% VHSV glycoprotein-expressing *C*. *vulgaris*. Furthermore, relative survival rates of 100% and 81.9% were achieved following challenges of these fish with VHSV at 10^6^ and 10^7^ pfu/fish, respectively, indicating that *C*. *vulgaris* could be used as a platform for the production of recombinant proteins for use as oral vaccines in the control of viral diseases in aquaculture.

## 1. Introduction

There is increasing demand for aquaculture, with fish consumption expected to remain high until 2030 [1,2]. However, the level of production has tended to decline compared with demand [2] due to the high cost of fish feed and disease outbreaks [3,4]. Feed accounts for more than 50% of all costs in the fish-farming industry [5,6]. To resolve the high cost of feed, microalgae have recently been used as feed additives instead of fishmeal [7,8]. Microalgae are rich in nutrients such as proteins, lipids, polysaccharides, and essential amino acids [7,9]. In addition, they can be cultured with sunlight as an energy source, grow quickly, function as an immune enhancer and probiotic when added to feed, and show sufficient digestion efficiency in various fish species [9,10].

Infectious diseases account for approximately 10% of losses in fish farming, mainly due to bacteria, parasites, and viruses [11,12,13]. Vaccines against fish diseases are commercially available in Korea, China, Japan, Europe, and North America, but most are directed against bacterial diseases and few have been developed for viral diseases [14,15,16,17]. Vaccines against fish viruses are urgently required [18].

Viral hemorrhagic septicemia (VHS) is a viral disease that seriously affects a variety of marine and freshwater fish, including salmonids and halibut, worldwide [19,20]. VHS is caused by viral hemorrhagic septicemia virus (VHSV), a single-stranded negative-strand RNA virus belonging to the genus *Novirhabdovirus* in the Rhabdoviridae family [21]. VHSV encodes six structural proteins: nucleoprotein (N), polymerase-associated phosphoprotein (P), matrix protein (M), glycoprotein (G), non-virion protein (NV), and large RNA-dependent RNA polymerase (L) [21]. There are four main viral genotypes (I, II, III, and IV), which are correlated with a wide range of geographic locations [22,23]. VHSV is listed as a notifiable pathogen by the World Organization for Animal Health (WOAH), and is highly contagious to a variety of fish species with high rates of morbidity [21,24]. It has been isolated from more than 80 species of fish, and at least 40 fish species are highly susceptible to the virus with adverse effects on aquaculture [23]. Despite these risks, very few vaccines are commercially available worldwide [15,25].

Fish vaccines are administered via intraperitoneal injection, intramuscular injection, immersion, and orally [15]. The first two routes show the greatest efficacy in fish but are sometimes associated with side effects, such as inflammation and organ adhesions [26,27]. Oral and immersion vaccines have fewer side effects than injected vaccines, but confer less protection [28]. Nevertheless, oral vaccines have the advantage of being less stressful and reducing the possibility of secondary infection, and so have been widely applied in fish vaccine development [29].

As mentioned above, microalgae are safe for use as feed additives for fish and can rapidly accumulate biomass [7,30]. In addition, they are useful platforms for producing complex structures or disulfide-rich recombinant proteins because they are capable of posttranslational modification and glycosylation [31,32,33]. Despite these many advantages, the levels of recombinant protein production in these platforms are not high [34,35]. Previously, we developed a salt-inducible promoter (SIP) and confirmed its activation via high NaCl concentrations. We transformed *Chlorella vulgaris* with a construct capable of expressing recombinant VHSV glycoprotein at high levels under control of the SIP, and investigated its efficacy as an oral vaccine against VHSV.

## 2. Materials and Methods

### 2.1. Culture Conditions for Chlorella vulgaris

The green microalga *C*. *vulgaris* PKVL7422 (KCTC13361BP) was cultivated in mBG11 medium. Cells were cultured at 20 °C with an incident light intensity of 52 μmol photons/m^2^/s^1^, and subcultured every 14 days.

### 2.2. Construction of Chlorella Transformation Vector

A *Chlorella* transformation vector containing the SIP (GenBank accession number OQ67458) from *C*. *vulgaris* PKVL7422 was constructed following a previously developed pCCVG vector as shown in Figure 1, with some modifications. pCCVG has two flanking sequences of 1000 bp derived from the *NR* gene of *C*. *vulgaris* PKVL7422 for integration via double homologous recombination encompassing the cauliflower mosaic virus (CaMV) 35S promoter, VHSV glycoprotein gene (GenBank accession number KM926343.1), and transcriptional terminator from the *Rbcs2* gene of *Chlamydomonas reinhardtii*. Both pCCVG and pSIP-T carrying the SIP were digested with *Bam*HI and *Eco*RI, and the CaMV 35S promoter in pCCVG was substituted with the SIP to construct pCSVG (Figure 1).

### 2.3. Chlorella Transformation

To secure a DNA fragment for homologous recombination, a 5013 bp DNA fragment was amplified using NR primers (Table 1). PCR was performed in LF™ 5× PCR Master mix (Elpisbio, Daejeon, Korea) with denaturation at 95 °C for 3 min followed by 35 cycles of amplification at 95 °C for 20 s, 59 °C 20 s, and 72 °C for 4 min with a final extension at 72 °C for 10 min. The PCR product was purified using electrophoresis on 1% (*w*/*v*) agarose gels followed by purification with a DokDo-Prep Gel/PCR Purification Kit (Elpisbio). Aliquots of 2 mL *C*. *vulgaris* PKVL7422 grown in BG11 medium to a cell count of 2 × 10^7^ cells/mL were harvested via centrifugation at 3000× *g* for 10 min at 4 °C. The pellets were resuspended in 400 µL osmosis buffer (200 mM d-sorbitol, 200 mM d-mannitol) and incubated for 1 h at room temperature. After centrifugation at 3000× *g* for 10 min, the pellets were resuspended in 400 µL electroporation buffer (500 mM NaCl, 200 mM d-mannitol, 200 mM d-sorbitol, 20 mM HEPES, 5 mM CaCl_2_, 5 mM KCl, pH 7.2) followed by addition of 4 µg DNA fragment and incubated on ice for 10 min. The mixtures were transferred to 0.2 cm electroporation cuvettes and electroporated at 1.0 kV, 25 μF, and 400 Ω for 5 s (Pulse Controller Plus; Bio-Rad, Hercules, CA, USA). Treated cells were transferred to 6-well plates containing 5 mL BGNK-broth [36] and stabilized in the dark at 20 °C for 18 h before plating on BGNK-agar plates [36]. The plates were incubated in the dark at 20 °C for 14 days.

### 2.4. Selection and Confirmation of Transformed Cells

After 14 days of incubation, 12 colonies of transformed cells were randomly selected, inoculated in 5 mL BGPK broth, and cultured for 7 days at 20 °C under fluorescent light at 52 μmol photons/m^2^/s^1^. DNA was extracted using a plant DNA extraction kit (NEH, Washington, DC, USA) according to the manufacturer’s instructions, and the VHSV glycoprotein gene was amplified with vG primers (Table 1).

### 2.5. Detection of Expressed Protein by Western Blotting

Samples (10 mL) of transformed *C*. *vulgaris* were cultured in BGPK broth to 2 × 10^8^ cells/mL and harvested via centrifugation at 3000× *g* for 10 min at 4 °C. The pellets were resuspended in 1000 µL RIPA buffer (Bio Basic, Markham, ON, Canada) and incubated on ice for 30 min. The cells were sonicated using a JY92-IIN Sonicator (Ningbo Scientz Biotechnology, Ningbo, China) equipped with a tip 2 mm in diameter and output of 200 W for 10 min (150 cycles of 2 s sonication with a 2 s pause). After centrifugation at 15,000× *g* for 2 min, the supernatant was transferred to a fresh tube. The concentration of total protein was measured using a Coomassie Plus (Bradford) Assay Kit (Thermo Fisher, Waltham, MA, USA) according to the manufacturer’s instructions. The extracted proteins were separated via sodium dodecyl sulfate-polyacrylamide gel electrophoresis (SDS-PAGE) at 110 V for 120 min, and were transferred onto PVDF membranes at 100 V for 120 min. Western blotting analysis was performed using anti-VHSV glycoprotein rabbit whole serum primary antibody (ABFrontier, Seoul, Republic of Korea) and anti-rabbit IgG(H+L)-HRP secondary antibody (Bio-Rad). The PVDF membranes were developed with WESTSAVE STAR™ (Young in Frontier, Seoul, Republic of Korea) for 1 min and analyzed with an iBright™ CL1500 imaging system (Thermo Fisher).

### 2.6. Analysis of Growth of Transformed Chlorella vulgaris According to NaCl Concentration

The growth rate of the transformed *C*. *vulgaris* was measured according to the NaCl concentration. NaCl was added to BGPK medium to final concentrations of 0 mM, 150 mM, 200 mM, 250 mM, 300 mM, 350 mM, 400 mM, and 450 mM. The initial cell concentration was 1 × 10^6^ cells/mL, and cells were incubated for 7 days at 20 °C under fluorescent light at 52 μmol photons/m^2^/s^1^. Wild-type *C*. *vulgaris* without NaCl treatment was used as a control. Cell number was counted every day using a hemocytometer with three replicates for each sample.

### 2.7. Induction of Protein Expression by NaCl Treatment

To examine the induction of recombinant VHSV glycoprotein expression, various concentrations of NaCl were added in the early log phase (long-term induction) or late log phase (short-term induction) according to the *Chlorella* growth curve. Transformed *C*. *vulgaris* was inoculated into BGPK broth with 200 mM KClO_3_ at an initial concentration of 1.5 × 10^6^ cells/mL. For long-term induction, cells were cultured for 2 days and then treated at the same concentrations of NaCl mentioned above. After incubation for an additional 5 days, protein was extracted and Western blotting analysis was performed. For short-term induction, cells were cultured for 7 days and NaCl was added at those same concentrations. After 30, 60, or 120 min, total protein was extracted and Western blotting was performed as described above. The expression level was compared via density analysis using iBright software (Thermo Fisher). The effects of two induction variables, i.e., NaCl concentration and treatment time, on the expression of VHSV glycoprotein in transformed *Chlorella* were evaluated using Design Expert 13 software (Stat-Ease Inc., Minneapolis, MN, USA).

### 2.8. Vaccination

For feed preparation, cultured *C*. *vulgaris* cells were centrifuged for 30 min at 3000× *g*. Samples of 10 g cell pellets were resuspended in 10 mL PBS and sonicated (JY92-IIN Sonicator; Ningbo Scientz Biotechnology), equipped with a tip 6 mm in diameter and output of 400 W for 10 min (150 cycles of 2 s sonication with a 2 s pause), and then freeze-dried (FDU-2200; Sunil Eyela, Seongnam, Republic of Korea) for 3 days. The resulting freeze-dried cells of about 3.3 g were resuspended with 40 mL PBS, sprayed onto 100 g feed (Neptune One; SAJO Dongaone Co., Ltd., Seoul, Republic of Korea), and allowed to dry for 3–4 days in the dark. Prepared feed was kept at −20 °C until use. Feed for the wild-type group was prepared by spraying wild-type *C*. *vulgaris* PKVL7422 onto the normal feed. Feed for the vaccine group were prepared by spraying transformed *C*. *vulgaris* PKVL7422 corresponding to 0.36 mg VHSV glycoprotein per 1 g feed (wet weight). The control group received normal feed containing phosphate-buffered saline (PBS).

Olive flounder fingerlings (*Paralichthys olivaceus*) with an average length of 16.2 cm and an average weight of 41.3 g were purchased from Chloland Co. Ltd. (Geoje, Republic of Korea) and confirmed to be free of pathogens, including VHSV. The fingerlings were divided into three groups with two replicates each of 20 fish per replicate in 150 L tanks. The fingerlings were fed with prepared feed to 3% of weight daily.

In the first week, test feed was given for 5 days followed by normal feed for 9 days. In the third week, test feed was again provided for 5 days followed by normal feed for 9 days and during the challenge period of 2 weeks.

### 2.9. VHSV Challenge

After 4 weeks of feeding, serum was obtained from five randomly selected fish from each group, and the remaining experimental fish were divided into two groups of 7–8 individuals each. The challenge experiment was conducted via intraperitoneal injection of two doses of wild-type VHSV KJ2008 strain (10^6^ and 10^7^ pfu/fish). The injected fish were kept in water at a temperature of 13 ± 1 °C for 15 days. The relative percentage survival (RPS) rate was confirmed by monitoring the cumulative mortality over time after VHSV infection and defined as follows:RPS = {1 – [mortality (%) of treated/mortality (%) of untreated control]} × 100

### 2.10. VHSV Neutralization Assay

Serum from three randomly selected individuals in each group was tested for antibody production and protective ability against VHSV infection using a virus neutralization assay. First, *epithelioma papulosum cyprini* (EPC) cells were seeded at 1 × 10^5^ cells/well in 96-well plates, cultured in Leibovitz’s L-15 medium (Thermo Fisher) containing 10% fetal bovine serum (Thermo Fisher) and antibiotic/antimycotic solution (Thermo Fisher) for 1 day at 28 °C, and then incubated at 20 °C for 1 day. Serum from sampled individuals was heated at 56 °C for 30 min to inactivate complement, and fresh serum (serum from flounder without any treatment) was diluted 1:250 without heat treatment. The serum samples from the experimental group were diluted from 2^–1^ to 2^–8^, and 50 μL diluted serum was mixed with 50 μL of fresh serum for complements and 50 μL of wild-type VHSV at 1 × 10^2^ pfu, followed by incubation at 15 °C for 24 h. After removing the medium from the cell culture, 100 μL of each sample was added to an EPC cell monolayer. The cells were incubated at 15 °C and the cytopathic effect (CPE) was observed every day. The titer of each serum sample was defined as the last dilution at which CPE was not observed.

### 2.11. Statistical Analysis

Data were analyzed with one-way and two-way repeated measures (RM) analysis of variance (ANOVA) using GraphPad Prism (GraphPad, San Diego, CA, USA) and the multiple *t* test using Microsoft Excel (Microsoft, Redmond, WA, USA). In all analyses, *p* < 0.001 was taken to indicate statistical significance.

## 3. Results

### 3.1. Transformation of Chlorella vulgaris

*C*. *vulgaris* PKVL7422 was transformed with the amplified DNA fragment and selected on BGNK plates. When cultured in the dark for 2 weeks, the transformed *C*. *vulgaris* formed colonies on BGNK cells, while wild-type *C*. *vulgaris* showed no colony formation on BGNK (Figure 2). Insertion of the VHSV glycoprotein gene was confirmed with PCR, which yielded an amplicon of the expected size of 559 bp with the vG primers (Table 1) specific to the VHSV glycoprotein in transformed *C*. *vulgaris*. Western blotting analysis was performed in the selected colonies and expression of VHSV glycoprotein (66.3 kDa) was confirmed in 4 of the 12 colonies examined, and one clone showing high expression was selected for NaCl induction analysis.

### 3.2. Growth of Transformed Chlorella vulgaris in Media Containing NaCl

The growth rates of transformed *C*. *vulgaris* at various NaCl concentrations were measured via cell counting. As shown in Figure 3, there were no significant differences in growth between wild-type and transformed *C*. *vulgaris* in cultures without NaCl. The growth of transformed *C*. *vulgaris* decreased with increasing NaCl concentration in the medium in a dose-dependent manner. The cell number of transformed *C*. *vulgaris* decreased to 50% in the presence of 300 mM NaCl compared with untreated control and decreased to only 15.8% at 450 mM NaCl (Figure 3).

### 3.3. Induction of Target Protein Expression by NaCl Treatment

As the VHSV G gene was cloned downstream of the SIP, which can be induced by NaCl treatment, the expression of the VHSV glycoprotein was analyzed at different NaCl concentrations and treatment times. The results of short-term treatment, in which cells were treated with different concentrations of NaCl for 30, 60, and 120 min before harvesting, are shown in Figure 4. In the case of 30 min treatment, the VHSV glycoprotein expression level increased with increasing NaCl concentration, and the highest expression level in the presence of 400 and 450 mM NaCl was 4.1 times higher than in the untreated control. VHSV glycoprotein expression in cells treated for 60 and 120 min before harvesting showed very similar patterns at the same NaCl concentrations, with the highest expression (3.3-fold increase over the control) in the presence of 200 mM NaCl decreasing gradually with further increases in NaCl concentration.

In long-term treatment, transformed *C*. *vulgaris* was cultured for 5 days in culture medium containing different concentrations of NaCl. The expression level was higher than that for the control in all NaCl-containing media, with the highest expression (9.6-fold increase over that of the control) in the presence of 250 mM NaCl, which decreased slightly at higher concentrations (Figure 4). Although the expression level was higher in the presence of a high NaCl concentration, cell growth was decreased at high NaCl levels. Therefore, transformed *C*. *vulgaris* was cultured in the presence of 250 mM NaCl in further experiments.

### 3.4. Optimization of Induction Parameters

We confirmed that the induction parameters had a significant influence on the level of recombinant protein expression (ANOVA, *p* < 0.0001). The fit of the model was good with an *R*^2^ value of 0.8199. Based on these results, the relations between VHSV glycoprotein production and two induction variables can be explained using the following quadratic equation, where A is time of day and B is NaCl concentration (in mM):Recombinant protein level = 4.91 + 1.87A + 1.49B + 1.02AB—1.92B^2^

Figure 5 shows 3D surface plots and 2D contour graphs as graphical representations of the combined effects of salt concentration and time on VHSV glycoprotein expression. In the long-term treatment, the level of VHSV glycoprotein expression was six times higher in the presence of 170–450 mM NaCl compared with that in the untreated control.

### 3.5. Quantification of VHSV Glycoprotein Expression

The amount of total protein extracted from 1 g wet transformed *C*. *vulgaris* cultured in BGPK medium with 250 mM NaCl was 15 mg. The amount of VHSV glycoprotein determined via density analysis and calculated using the quadratic equation outlined above was 4.11 mg/g wet transformed *C*. *vulgaris*, corresponding to 27.4% of total soluble protein (TSP).

### 3.6. Vaccine Efficacy of VHSV Glycoprotein-Expressing Chlorella vulgaris

Induction of an immune response by VHSV glycoprotein provided by feed was confirmed via neutralization of virus infection using serum from vaccinated fish. As shown in Figure 6, the serum from fish fed transformed *C*. *vulgaris* expressing VHSV glycoprotein showed significant inhibition of VHSV infection compared with that from controls and fish fed wild-type *C*. *vulgaris*.

Following VHSV challenge at 10^6^ pfu/fish, the cumulative mortality rates of the control group and fish fed wild-type *C*. *vulgaris* were 26.8% and 6.25%, respectively, after 15 days (Figure 6). However, no fish died due to infection in the group fed VHSV glycoprotein-expressing transformed *C*. *vulgaris*. In the case of VHSV challenge at 10^7^ pfu/fish, the cumulative mortality rates of the control group and fish fed wild-type *C*. *vulgaris* were 78.6% and 35.7%, respectively, after 15 days. On the other hand, fish fed transformed *C*. *vulgaris* expressing VHSV glycoprotein had a cumulative mortality rate of 14% with an RPS of 81.9%.

## 4. Discussion

Inactivated vaccines, DNA vaccines, and recombinant protein vaccines are effective for preventing VHSV [37,38,39]. A mixed DNA vaccine encoding VHSV glycoprotein delivered via intramuscular injection is effective for protecting against the virus for up to 80 days [40]. Oral vaccines are preferred over injected vaccines due to safety issues as, despite their high protective efficiency, injected vaccines may result in lesions, such as organ adhesions, and secondary infection [41,42].

Oral vaccines are prepared by coating or mixing antigens, such as recombinant proteins, with feed, and have the advantage of being less stressful to fish [43]. Microalgae, nanoparticles, bacteria, and plant-based proteins have been used as oral vaccine materials [43].

*C*. *vulgaris*, which contains approximately 61.6% protein, 12.5% fat, and 13.7% carbohydrates and is rich in nutrients, such as minerals and vitamins, has been used as a growth promoter and immune enhancer for fish [44]. *Chlorella* and other microalgae have been developed as industrial hosts for recombinant protein production [45]. However, microalgal expression systems have the considerable drawback of low expression levels of exogenous proteins [45]. For example, in previous studies, the expression level of recombinant basic fibroblast growth factor (bFGF) was 1.61 μg/kg in *C*. *vulgaris* compared with 42 g/kg in *E*. *coli*, and 185.7 mg/kg in transgenic rice seeds [45,46,47]. In other studies, the expression level of viral surface protein 2 (VP2) of infectious pancreatic necrosis virus in transformed *Nannochloropsis oceanica* was 1–4.9% of TSP, and those of recombinant proteins in *Chlamydomonas reinhardtii* were reported to range from 0.1% to 10% of TSP [48,49].

To increase the expression level of recombinant protein, *C*. *vulgaris* expressing the VHSV glycoprotein was developed using the SIP developed in a previous study [50]. Although the yield of recombinant protein can be increased with NaCl treatment, the growth of transformed *C*. *vulgaris* was also affected by NaCl concentration because this is a freshwater *Chlorella* strain. Therefore, the growth of transformed *C*. *vulgaris* was measured at different NaCl concentrations. As shown in Figure 3, the growth of *C*. *vulgaris* decreased with increasing NaCl concentrations. However, even at the highest concentration of 450 mM, the final cell count after 7 days was 16.8 times the initial value, indicating that it could survive even in the presence of high salt concentrations due to its response and adaptability to abiotic stress [51,52].

To determine the optimum induction conditions for maximum yield of the VHSV glycoprotein, we examined two different induction methods involving short-term and long-term treatment with different NaCl concentrations. In short-term treatment, cells were cultured for 7 days to the stationary phase and NaCl was added 30, 60, or 120 min before harvesting. Treatment for 30 min increased the expression of VHSV glycoprotein in a dose-dependent manner (Figure 4). However, the expression level was highest at 200 mM with both 60 and 120 min treatments and decreased at higher concentrations. Similarly, the accumulation of cTDA1-mediated recombinant green fluorescent protein (GFP) expressed after heat shock in *Chlamydomonas reinhardtii* was highest at 30 min but decreased again after 1 to 2 h, indicating that the expression rate differed over time [53].

In the long-term treatment experiment in which cells were treated with different concentrations of NaCl for 5 days before harvesting, VHSV glycoprotein expression level increased up to 250 mM NaCl (9.6-fold higher than that in the untreated control). Ruecker et al. [54] also reported that the level of *Gaussia* luciferase expression under control of the heat-inducible HSP70A promoter in transformed *C*. *reinhardtii* was highest at 250 mM NaCl.

The protein content of *C*. *vulgaris* decreased with increasing NaCl concentration due to the denaturation of chlorophyll, membrane damage, and inhibition of photosynthesis [55,56]. In our study, NaCl concentration resulted in growth inhibition in a dose-dependent manner (Figure 3). Analyses using 3D surface plots and 2D contour graphs showed significant correlations among these factors (Figure 5). These observations and the results of Western blotting analysis showed that 250 mM NaCl and long-term treatment were optimal conditions for maximizing the yield of VHSV glycoprotein. Under these conditions, 41.1 mg recombinant VHSV glycoprotein per 10 g wet weight of transformed *C*. *vulgaris* in 1 L culture was produced, representing 27.4% of the TSP. The yields of recombinant proteins from transformed microalgae were reported to range from 0.006 to 6.4 mg/L, corresponding to 0.1% to 10% of the TSP [45,48,49]. Therefore, the expression level achieved here by NaCl induction was efficient for the production of recombinant vaccine protein and may also be applicable to the production of other valuable recombinant proteins in the future.

The induction of an immune response to VHSV glycoprotein was first examined using a neutralization assay with serum from fish given feed containing transformed VHSV glycoprotein-expressing *C*. *vulgaris*. Although background neutralization was detected with serum from fish given wild-type *C*. *vulgaris*, significant neutralization was observed in fish given feed containing transformed *C*. *vulgaris*. Western blotting analysis with VHSV lysate as the antigen and olive flounder serum (1:500), rabbit anti-olive flounder IgM (1:1000), and alkaline phosphatase-conjugated goat anti-rabbit IgG (1:2000) as the primary, secondary, and tertiary antibodies, respectively, detected the production of IgM antibody to VHSV glycoprotein.

In general, the survival of vaccinated fish after challenge with VHSV is highly correlated with the neutralizing activity in the serum [57]. The efficacy of an orally provided vaccine was examined by challenging immunized fish (Figure 6). A wide range of doses from 10 μg to 100 μg have been used for intramuscular vaccination with recombinant subunit vaccines [58,59,60]. However, there is insufficient information regarding the efficacy of oral vaccination against VHSV. Doses of 10^8^–10^10^ cfu/g of *Lactococcus lactis* expressing VHSV glycoprotein [37] and 10^5^–10^10^ 50% tissue culture infectious dose (TCID_50_)/fish of inactivated recombinant virus have been used for oral vaccination [61]. The addition of 5–15% *C*. *vulgaris* to the feed increased the growth of fish by up to 55% and further increased the antioxidant effects [58,61]. In preliminary experiments, we added 5% transformed *C*. *vulgaris* to normal feed. However, there was feed left over, so we increased the *C*. *vulgaris* content to 9% and reduced the amount of feed provided, which resulted in a dose of 4.5 mg VHSV glycoprotein/fish during the 10-day oral vaccination period.

In addition to the vaccine dose, virus titer can affect the outcome of a challenge test. Depending on species and size, virus titers from 10^4^ to 10^7^ pfu/fish have been used in VHSV challenges [60,62]. In this study, two different titers of VHSV, 10^6^ and 10^7^ pfu/fish, were used in the challenge, and higher mortality was observed at the higher titer. Fish given feed containing wild-type *C*. *vulgaris* showed lower mortality rates than the control group given normal feed upon VHSV challenge with virus titers of both 10^6^ pfu/fish (6.3% vs. 26.8%, respectively) and 10^7^ pfu/fish (35.7% vs. 78.6%, respectively). Extracts of some species of microalgae, including *Chlorella* species, have antiviral effects [63], and *C*. *vulgaris* has been shown to improve growth, immunity, and antioxidant properties when used as a feed additive [64]. Therefore, *C*. *vulgaris* used as the industrial host for VHSV glycoprotein production could act as an immunostimulant in oral vaccination. This was further confirmed by the cumulative mortality rates of fish fed with VHSV glycoprotein-expressing *Chlorella*, which were 0% and 14.0% on challenge with virus titers of 10^6^ pfu/fish and 10^7^ pfu/fish, respectively, corresponding to RPS of 100% and 81.9%, respectively. Naderi-Samani et al. [37] reported an RPS of 78% obtained by oral provision of *Lactococcus lactis* expressing VHSV glycoprotein. In addition, Kole et al. [65] reported an RPS of 0% and 20% following oral vaccination with inactivated and nanoencapsulated VHSV, respectively, on 2 consecutive days (twice/day) with the same booster immunization 2 weeks after primary immunization. Although the amount of VHSV glycoprotein given as vaccine and the methods used in challenge tests were slightly different between these studies and the present study, our findings indicate the protective effects of VHSV glycoprotein expressed in *C*. *vulgaris* as an oral vaccine against VHSV infection in a major aquaculture fish species.

## 5. Conclusions

Considering the immunostimulant effect of *C*. *vulgaris* itself, the high level of VHSV glycoprotein expression obtained by the SIP used in this study, the low cost of culture with high cell mass production, and the protective effect of oral vaccination, the system described here can be applied for the production of oral vaccines for VHSV and other viruses responsible for significant economic losses in aquaculture.

## Figures and Tables

**Figure 1 vaccines-11-01205-f001:**
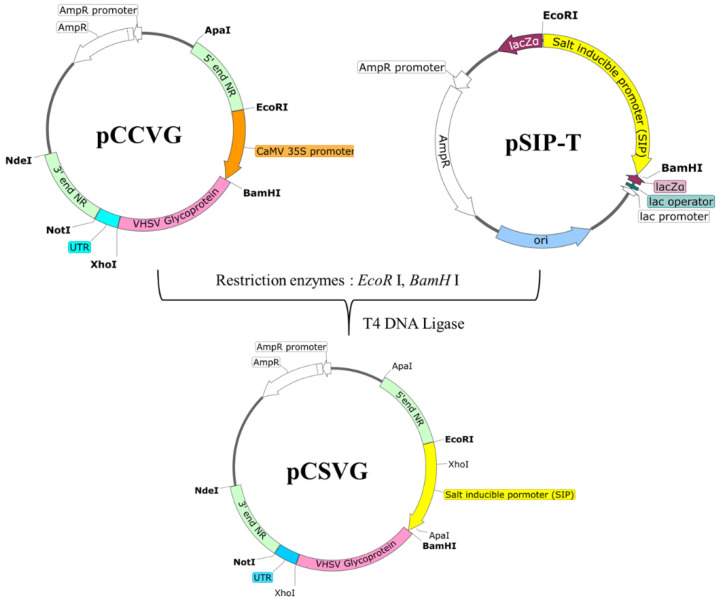
Construction of *Chlorella* transformation vector for VHSV glycoprotein expression. The CaMV 35S promoter in pCCVG was replaced with the salt-inducible promoter (SIP) from *C*. *vulgaris* PKVL7422, yielding the pCSVG vector.

**Figure 2 vaccines-11-01205-f002:**
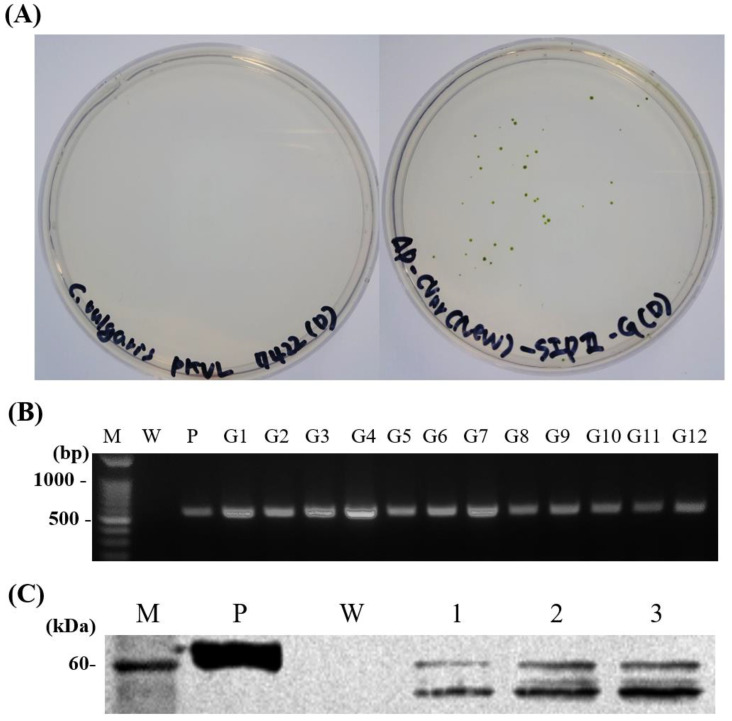
Selection and confirmation of transformant expressing VHSV glycoprotein. (**A**) Selection of transformants on BGNK plates in the dark: left panel, wild-type *C*. *vulgaris* PKVL7422; right panel, transformed *C*. *vulgaris*. (**B**) Detection of DNA insert with PCR analysis of selected transformed cells. Lane M, DM3200 DNA marker (SMOBiO Technology, Hsinchu, China); lane W, wild-type *C*. *vulgaris* PKVL7422; lane P, positive control (plasmid vector); lanes G1–G12, transformed *C*. *vulgaris* (SIPG). (**C**) Western blotting analysis of selected transformed cells. Lane PM, PM2700 protein marker (SMOBiO); lane V, purified VHSV (positive control); lane W, wild-type *C*. *vulgaris* PKVL7422; lanes 1–3, transformed *C*. *vulgaris*.

**Figure 3 vaccines-11-01205-f003:**
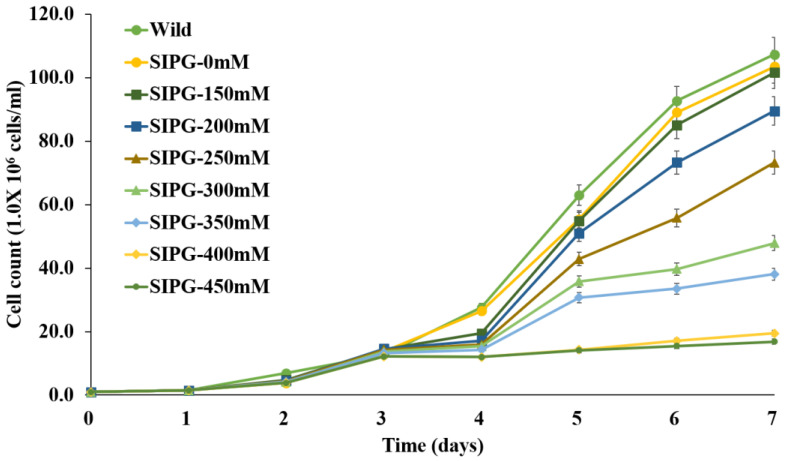
Effects of NaCl concentration on growth of transformed *C*. *vulgaris*. Transformed *C*. *vulgaris* was cultured in medium containing different concentrations of NaCl and growth was measured using direct cell counting. Wild-type *C*. *vulgaris* PKVL7422 (Wild) cultured in mBG11 medium without NaCl was used as a control.

**Figure 4 vaccines-11-01205-f004:**
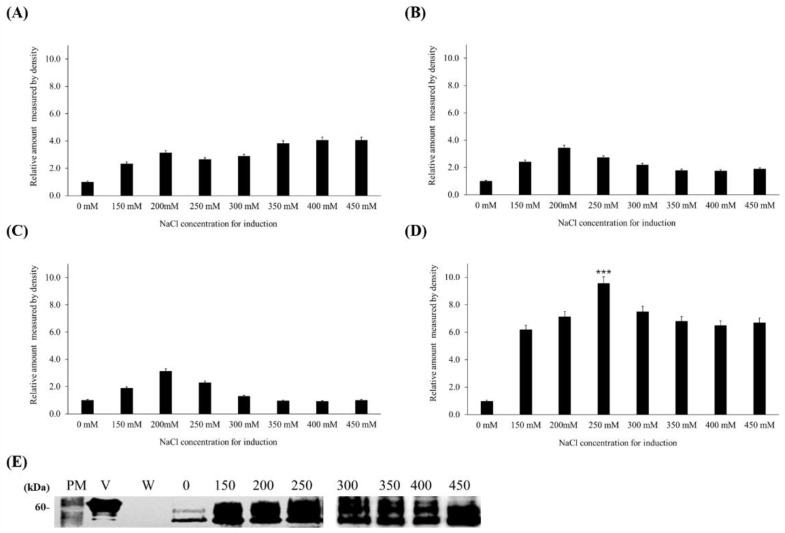
Comparison of VHSV glycoprotein expression according to treatment time and NaCl concentration. (**A**–**C**) Relative expression levels of VHSV glycoprotein in cells treated for 30 min (**A**), 60 min (**B**), and 120 min (**C**) with different concentrations of NaCl. (**D**) Relative expression level of VHSV glycoprotein in cells cultured for 5 days with different concentrations of NaCl. (**E**) Western blotting analysis for quantification of the expression levels in (**D**). Lane PM, PM2700 protein marker; lane V, purified VHSV (positive control); lane W, wild-type *C*. *vulgaris* PKVL7422; lane 0–450, transformed *C*. *vulgaris* cultured in the presence of the corresponding NaCl concentration (in mM). The expression level at 250 mM that was significantly different from those at other concentrations with *p* < 0.0001 is marked with ***.

**Figure 5 vaccines-11-01205-f005:**
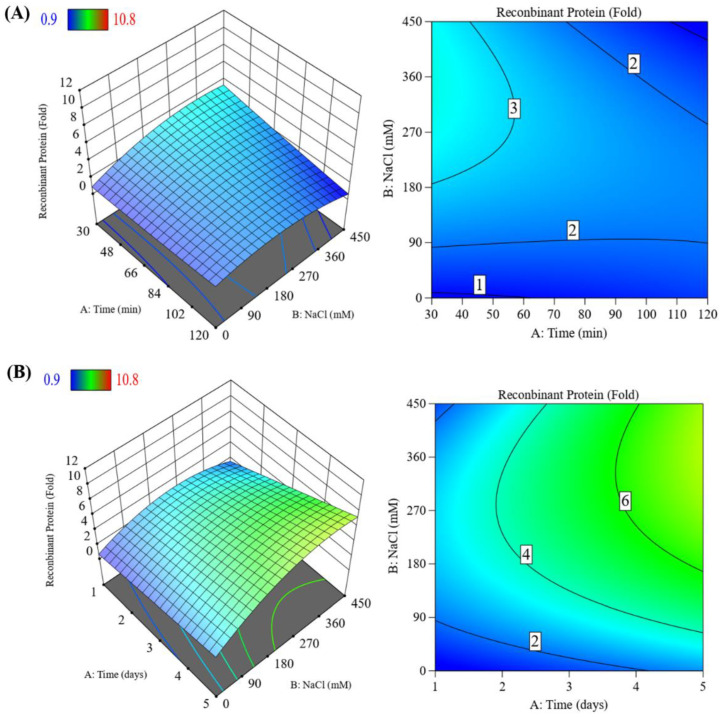
Three-dimensional surface plots and two-dimensional contour graphs showing relative VHSV glycoprotein expression levels. (**A**) Three-dimensional and two-dimensional plots showing the induction of VHSV glycoprotein expression with short-term treatment at different NaCl concentrations. (**B**) Three-dimensional and two-dimensional plots showing the induction of VHSV glycoprotein expression with long-term treatment at different NaCl concentrations. The numbers in the boxes indicate fold increases in expression level compared with untreated control.

**Figure 6 vaccines-11-01205-f006:**
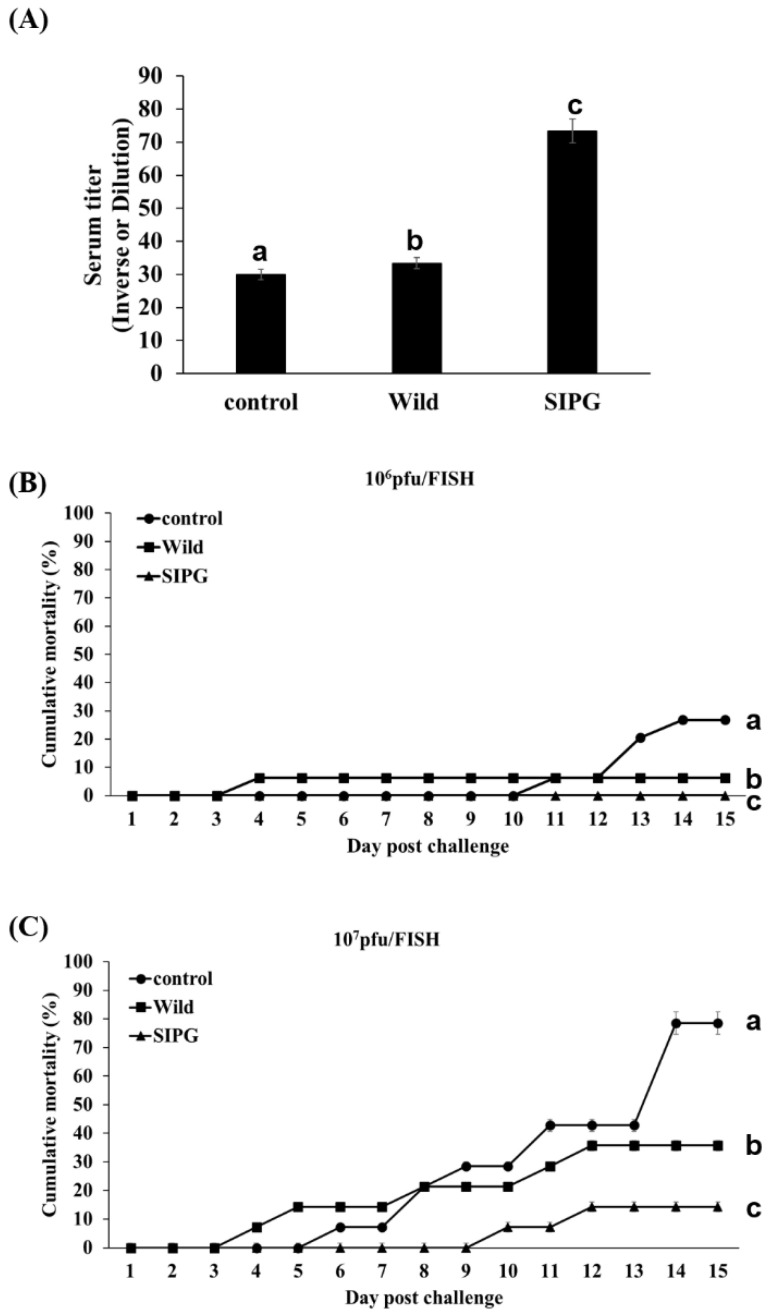
Efficacy of transformed *C*. *vulgaris* expressing VHSV glycoprotein as an oral vaccine. (**A**) Inverse dilution of serum titer for neutralization with serum from fish given normal feed (Control), feed containing wild-type *C*. *vulgaris* (Wild), and feed containing transformed *C*. *vulgaris* expressing VHSV glycoprotein (SIPG). (**B**) Cumulative mortality rates of fish fed as in (**A**) and challenged with 10^6^ pfu/fish. (**C**) Cumulative mortality rates of fish fed as in (**A**) and challenged with 10^7^ pfu/fish. Different letters indicate statistically significantly differences (*p* < 0.001).

**Table 1 vaccines-11-01205-t001:** Primers used.

Target	Primer Name	Sequences	PCR Product
NR	NRF	5′-ATGGACAAGACAGGGTTCGG-3′	5013 bp
	NRR	5′-AATACAGGCGGAGCCCAAAC-3′	
vG	vGF	5′-CAATTTCAGAAAGAATGCTAACC-3′	559 bp
	vGR	5′-CACTATCTTCACAATAAAGTGAC-3′	

## Data Availability

The data presented in this study are available on request from the corresponding author. The data are not publicly available due to the agreement of technical transfer.

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
