# Peer review of "High-Level Expression of Recombinant VHSV Glycoprotein Using Transformed C. vulgaris and Verification of Vaccine Efficacy"

_vaccines, 2023, doi:10.3390/vaccines11071205_

Round 1
Reviewer 1 Report
The MS is very interesting research in the field of vaccine development with advanced molecular biology tools. The MS is well planned, organized and also analyzed the data is a good manner. The research is solved many demerits of normal vaccine delivery. Conventional vaccine delivery has many drawbacks like painful and loss of antigens etc. The delivery through micro algae have many advantages like safety and ecofriendly, also deliver to post larvae also. the small fishes have easily feed the micro algae with recombinant vaccines. I strongly recommended the MS for accepting.
Author Response
Thank you for your kind review and comments
Reviewer 2 Report
In sub-heading Vaccination- line 3 authors used the word "fry" while the fish they used were 41 g thus fish at this size is fingerling not fry.
In sub-heading-VHSV neutralization assay
The serum samples from the experimental group were diluted from 2–1 to 2–8 , and 50 μL diluted serum was mixed with 50 μL fresh serum and 50 μL wild-type VHSV at 1 × 102 pfu followed by incubation at 15°C for 24 h. After removing the medium from the cell culture, 100 μL of each sample was added to a CPE cell monolayer.
1. Why used 50 μL fresh serum
2. What means by CPE cell monolyaer- perhaps you means ECP cell line.
3. Please clearly revise this paragraph and describe you NT clearly.
The language can be improved by a native English language.
Author Response
-
- Why used 50 μL fresh serum ?
- Fresh serum was added as a source for complement. The diluted samples have heat treated to inactivate the complement whose concentration is different in each dilution. So, fresh serum was added to provide the same amount of complement in all dilutions. The reason for using fresh serum is provided in the text.
- What means by CPE cell monolyaer- perhaps you means ECP cell line.
- Corrected as ECP
- Please clearly revise this paragraph and describe you NT clearly.
- This paragraph was written as follow with the answer to question number 1.
“ The serum samples from the experimental group were diluted from 2–1 to 2–8, and 50 μL diluted serum was mixed with 50 μL of fresh serum for complements and 50 μL of wild-type VHSV at 1 × 102 pfu, followed by incubation at 15°C for 24 h. After removing the medium from the cell culture, 100 μL of each sample was added to a EPC cell monolayer.”
Reviewer 3 Report
There is a need for improved oral vaccines in fish and this paper has an interesting approach to enabling a VHSV vaccine. The use of Chlorella vulgaris and their inclusion of a native C. vulgaris treatment (in addition to the recombinant G protein C. vulgaris) was very enlightening - showing value of the C. vulgaris alone. They synergy between the two was also evident. You did a good job in discussion of these results and the paper is in very good shape.
Figures 1, 4 & 6 need to be improved, since the writing is not legible. Figure 5 could also be sharper.
In Figure 4, can you show that 250 nM is significantly different than the other salt concentrations?
The incorporation of the recombinant C. vulgaris into the feed in the Methods section needs to be clarified. Switching from wet weight content to freeze dried broken cells that are top coated was confusing.
Overall very good paper.
Author Response
-
Figures 1, 4 & 6 need to be improved, since the writing is not legible. Figure 5 could also be sharper.
- The figures are replaced with improved image. Actually the figures were shaded during electric conversion of the original submission.
2. In Figure 4, can you show that 250 nM is significantly different than the other salt concentrations?
- High level expression at the p values of 0.0001 is marked in the figure legend.
3. The incorporation of the recombinant C. vulgaris into the feed in the Methods section needs to be clarified. Switching from wet weight content to freeze dried broken cells that are top coated was confusing.
- The whole paragraph is rewritten to clarify feed preparation with the same units as follow:
“ For feed preparation, cultured C. vulgaris cells were centrifuged for 30 min at 3000 × g. Samples of 10 g cell pellets were resuspended in 10 mL PBS and sonicated (JY92-IIN Sonicator; Ningbo Scientz Biotechnology) equipped with a tip 6 mm in diameter and output of 400 W for 10 min (150 cycles of 2 s sonication with a 2 s pause) and then freeze-dried (FDU-2200; Sunil Eyela, Seongnam, Korea) for 3 days. The resulting freeze-dried cells of about 3.3 g were resuspended with 40 mL PBS and sprayed onto 100g feed (Neptune One; SAJO Dongaone Co., Ltd., Seoul, Korea), and allowed to dry for 3–4 days in the dark. Prepared feed was kept at –20°C until use. Feed for the wild-type group were prepared by spraying wild-type C. vulgaris PKVL7422 to the normal feed. Feed for the vaccine group were prepared by spraying transformed C. vulgaris PKVL7422 corresponding to 0.36 mg VHSV glycoprotein per 1 g feed (wet weight). The control group received normal feed containing phosphate buffered saline (PBS).
Olive flounder fingerlings (Paralichthys olivaceus) with an average length of 16.2 cm and an average weight of 41.3 g were purchased from Chloland Co. Ltd. (Geoje, Korea) and confirmed to be free of pathogens, including VHSV. The fingerlings were divided into three groups with two replicates each of 20 fish per replicate in 150 L tanks. The fingerlings were fed with prepared feed to 3% of weight daily.
In the first week, test feed was given for 5 days followed by normal feed for 9 days. In the third week, test feed was again provided for 5 days followed by normal feed for 9 days and during the challenge period of 2 weeks.”